# Dimensions of aberrant driving behaviors and their association with road traffic injuries among drivers

Forouzan Rezapur-Shahkolai[1], Malihe Taheri[2], Tahereh Etesamifard[2], Ghodratollah Roshanaei[3]*, Samane Shirahmadi[2]*

1 Department of Public Health, School of Public Health and Social Determinants of Health Research Center, Hamadan University of Medical Sciences, Hamadan, Iran, 2 Department of Public Health, School of Public Health and Student Research Committee, Hamadan University of Medical Sciences, Hamadan, Iran, 3 Department of Biostatistics, School of Public Health and Modeling of Noncommunicable Diseases Research Center, Hamadan University of Medical Sciences, Hamadan, Iran

☯ These authors contributed equally to this work.
* s_shirahmadi@yahoo.com (SS); gh.roshanaei@umsha.ac.ir (GR)

## Abstract

### Objective

Road traffic injuries (RTIs) are recognized as one of the most important causes of morbidity and mortality throughout the world, especially in developing countries. Human behavior is reportedly one of the critical factors in the occurrence of such injuries. The purpose of this study is to evaluate the correlation of abnormal driving behaviors with the frequency and severity of RTIs among drivers in Hamadan, west of Iran.

### Methods

The present cross-sectional study was conducted on 800 people driving, who were selected by multistage cluster sampling technique. Data were collected using a three-part self-administered questionnaire including demographic, social and driving characteristics; the Manchester driver behavior questionnaire (DBQ); as well as information on a history of the occurrence of the injuries caused by the crashes and the severity of them. Data were statistically analyzed using numerical indices, linear regression analysis, Pearson correlation, ordinal logistic regression model and multinomial logistic regression.

### Results

The highest and lowest mean percentages of abnormal driving behavior were related to unintentional violations (19.13) and Lapses (16.44), respectively. "Changing radio stations and listening to music while driving", "overtaking a driver who drives slowly", and "unintentionally exceeding the speed limit" were the three highest behaviors associated with road traffic injuries, with the mean and standard deviation of (1.93 ± 1.4), (1.90±1.4), (1.58±1.3), respectively. Age, gender, educational level, driving experience and driving hours during the day were significantly associated with DBQ dimensions and severity of road traffic injuries.

**Data Availability Statement:** We have provided all relevant data within the paper and it's supporting information as a SPSS database file. The data are de-identified.

**Funding:** The authors who received award: Forouzan Rezapur-Shahkolai Grant numbers awarded to author: 930126254 The full name of funder: Deputy of Research and Technology of Hamadan University of Medical Sciences The funders had no role in study design, data collection and analysis, decision to publish, or preparation of the manuscript.

**Competing interests:** The authors have declared that no competing interests exist.

## Conclusions

The results of this study showed that socio-demographic characteristics were significantly correlated with driving behavior. In addition, driving behaviors were correlated with traffic crashes and the resulting injuries. The findings of this study can be utilized to develop driving behavior interventions among the drivers.

## Introduction

According to statistics, road traffic injuries (RTIs) are recognized as one of the most important causes of morbidity and mortality throughout the world, especially in developing countries [1].

According to the World Health Organization, the number of people killed due to RTIs continues to increase gradually, reaching 1.35 million in 2016 [2]. RTIs are currently the eighth leading cause of death in all age groups and the main cause of death in children and adults aged between 5 to 29 years, indicating inadequate efforts to achieve the sustainable development goal (SDG) to reduce road traffic deaths by 50% by 2020 [2].

The world's population is becoming increasingly urban, expansion of machine technology and development of cities have increased the use of cars [3, 4]. This has made driving a necessity for work, social life, leisure, educational and economic activities [5]. The increase in the need for driving led to an increase in traffic crashes and, consequently increased injuries and deaths, resulting in the loss of manpower and the burden of economic pressure on healthcare systems [3, 4]. Consequently, traffic injuries and deaths are recognized as serious public health problems [3, 4].

There is a direct correlation between the risk of death from traffic crashes and the income level of countries. The mean rate of traffic deaths in low-income countries (27.5 deaths per 100,000 people) is three times higher than in high-income countries (8.3 deaths per 100,000 people) [2].

While the global rate of traffic deaths is 18.2 per 100,000 people, there are significant differences across the globe. Regional mortality rates per 100,000 people, respectively, are 26.6 and 20.7 deaths in Africa and Southeast Asia, 18 and 16.9 deaths in the Eastern Mediterranean and Western Pacific, and 15.6 and 9.3 deaths in America and Europe. Adoption and enforcement of law on key behavioral risk factors has been an important component of integrated strategy to prevent traffic deaths in high-income countries [2].

Although traffic crashes are the leading cause of annual deaths, a key difference between Iran and other developing countries is the increasing trend in its traffic crashes rate index. Annually, more than 400,000 traffic crashes occur in Iran, 20 times higher than the global average rate, with injuries the fifth leading cause of death in the age group of less than 40 years and the second leading cause of potential years of life lost and disability [6]. The death rate in Iran is 30 deaths per 10000 vehicles while this rate is 1 to 2.5 deaths in developed countries and 3 to 15 deaths in developing countries [7]. According to a report from the World Health Organization Statistics in 2019, Iran's rate for road traffic injuries leading to death was 20.5 per 100000 population in 2016 [8].

In addition, the RTIs impose a burden of more than $ 4.4 billion annually on the Iranian economy, equivalent to 2.18% of gross domestic products (GDP). The mentioned burden only relates to the health sector and does not include items such as car damage, time lost in crashes, and administrative costs of insurance and police services. However, it is equivalent to 75% of the total budget allocated to the health sector [9, 10].

To date, considerable research efforts have been devoted to identify the factors affecting the traffic crash risk and severity. The results of these studies have shown that driving environment, spatial correlation and driving behavior affect the RTIs, traffic crash risk and severity [11–15]. Along with other factors affecting RTIs, driving behavior plays a key role in causing these injuries [16]. According to a report by the Traffic Police of Iran in 2014, human behavior (71%), technical defect (18%) and traffic engineering (11%) were in the traffic crashes [17]. Therefore, study of human behaviors among Iranian drivers should be considered very cautiously.

Driving skills and driving style represent two complementary and independent driving behaviors. The driving skills are those behaviors, increased or improved by practice, while driving behavior consists of driving habits developed over the years. The latter do not necessarily become safer over time and with driving experience [18].

Driving behaviors, which consist of two classes of lapse errors and violations, are the behaviors chosen by the driver to drive in a pattern such as driving speed, having concentration while driving and maintaining a standard distance [19, 20].

Reason et al. has divided abnormal driving behaviors into errors and violations and developed a driving behavior questionnaire to measure these concepts [21]. Errors are caused by cognitive processing problems and include "slips" and "lapses", while violations contain a motivational component and include deliberate and unintentional violations [22, 23].

Given the limited similar study in Iran and considering the impact of each country's specific cultural background and context on driving behaviors, this study aims to investigate abnormal driving behaviors and their association with RTIs.

## Materials and methods

### Ethical considerations

The Ethics Committee of Hamadan University of Medical Sciences approved this study (1930). All participants gave the informed consent after taking information about study objectives, being confidentially and being voluntary to participate in the study. All data were collected from the study participants anonymously. The participants in this study were not paid any direct benefits or rewards for their participation.

### Study design and population

This cross-sectional study was conducted in 2016 on people driving in the city of Hamadan (west of Iran).

### Sampling

The sample size was estimated at 800 according to a previous study [24] and considering sampling formula for cross-sectional study, with 95% confidence interval, maximum error significance level of 0.22 and clustering coefficient of 1.5 with regard to 10% dropout rate.

Considering the severity of RTIs [20], the participants who had crash experience in past year in this study were divided into four: 1- Drivers with traffic crash, but no injuries (vehicle damage only), 2- Drivers with traffic crash, resulting in mild injury (injuries requiring outpatient care or hospital stay less than 24 hours), 3- Drivers with traffic crash, resulting in moderate injury (injuries requiring hospital stay more than 24 hours) and 4- Drivers with traffic crash resulting in severe injury (fatal injuries). Later, the estimated sample size was calculated according to the size of each of the mentioned groups or classes based on the number of crashes in 2012 [25].

The sampling method was a multistage cluster sampling technique with the aim of selecting eligible participants. To collect the data, the city of Hamadan was first divided into four geographical districts: North, South, East and West, followed by selecting one neighborhood from each district, one block from each neighborhood, and 30 random households from each block. The questionnaire was introduced to a family member who had a driver's license with at least one year of driving experience. To distribute the self-administered questionnaires, the researchers referred to the first residential unit on the right side of the selected street and began collecting data and continued sampling until reaching the required sample size.

## Data collection tool

A three-part self-administered questionnaire was used for data collection as follows:

*Demographic, socioeconomic, and driving characteristics*:

The socio-demographic profiles of drivers and driving characteristics were collected in the first part of the questionnaire. Age, gender and educational level of the participants were recorded in nominal variables (high school and lower, associate degree/bachelor's degree, master's degree and higher).

Driving characteristics included the driving experience that counts the total number of years a person has driven a car (one year, 2–5 years, 6–10 years, and more than 10 years), driving time during the day in hours (less than 3 hours, 3–6 hours, and more than 6 hours) and people fined for driving over the past year (the participants were asked to choose one of the yes or no options for asking if "Have you been fined in the previous year?"

## Driver behavior questionnaire measurement

The standard localized Manchester driver behavior questionnaire (DBQ) designed to determine the extent of involvement of a variety of abnormal driving behaviors was used in this study. The tool consists of fifty questions in four domains of slips (n = 21), lapses (n = 9), deliberate violations (n = 17) and unintentional violations (n = 3) 2, and scored on a 6-point Likert scale ranging from 0 to 5 (never = 0, rarely = 1, occasionally = 2, often = 3, frequently = 4, always = 5). A score higher than mean value indicated the highest driver errors and violations in driving behaviors. The validity of the DBQ has been validated in numerous studies conducted in Iran [26–28].

## Road traffic crashes and injuries

The participants were asked about the history of RTIs, including the frequency and severity of these injuries.

We assessed the frequency of traffic crashes for drivers during the past year (1- I had no traffic crash, 2- I had one traffic crash, 3- I had two traffic crashes, 4- I had three traffic crashes, 5- I had four or more traffic crashes) and the severity of injuries due to traffic crashes (1- no injuries, 2- injuries requiring hospitalization less than 24 hours, 3- injuries requiring hospitalization more than 24 hours and 4- fatal injuries).

## Data analysis

Statistical analysis was performed using SPSS version 16 software. Descriptive statistics (mean, standard deviation, frequency and ratio) were calculated for all demographic variables and DBQ dimensions. The Pearson correlation coefficient was used to determine the correlation between DBQ dimensions.

Regarding the correlation between background variables and DBQ dimensions in several studies [29–31], four linear regression models were used to determine the correlation between

scores of DBQ dimensions and background variables (socio-demographic and driving variables).

Analysis of crash variable as ordinal response variable was performed using ordinal logistic regression model (proportional odds model (POM)).

To estimate the Ln (odds) of being at or below the j$^{th}$ category, the PO model can be rewritten as:

$$logit[\pi(Y \leq j \mid x_1, x_2, \ldots, x_p)] = ln\left(\frac{\pi(Y \leq j \mid x_1, x_2, \ldots, x_p)}{\pi(Y > j \mid x_1, x_2, \ldots, x_p)}\right)$$
$$= \alpha_j + (-\beta_1 X_1 - \beta_2 X_2 - \cdots - \beta_p X_p)$$

Thus, this model predicts cumulative logits J-1 response categories. By transforming the cumulative logits, we can obtain the estimated cumulative odds as well as the cumulative probabilities being at or below the j$^{th}$ category.

The chi-squared score test for the proportional odds assumption in ordinal logistic was employed to see whether the main model assumption was violated or not.

In this study ordered dependent variable consists of 5 ordinal categories (no traffic crash, one traffic crash, two traffic crashes, three traffic crashes, four or more traffic crashes). Also, multinomial logistic regression (multinomial regression) was used to predict a multinomial dependent variable (injury variable) given one or more independent variables. The 95% confidence interval (CI) and odds ratio were calculated. A Significance level of $P < 0.05$ was considered in all statistical tests.

## Results

The mean (±SD) age, driving experience, and hours of driving a day were 35.42 (±11.85) years, 10.57 (±9.26) years and 3.37 (±2.65) hours, respectively. Table 1 shows the characteristics of participant's driving behaviors, including lapse errors, slip errors, deliberate violations and unintentional violations according to their demographic variables. Based on this table, most participants were in the age range of 25 to 35 years (30.8). In addition, 85% of drivers were fined during the past year and 53.4% had a traffic crash at least once.

There was a significant correlation between DBQ dimensions (P = 0.001).

The means and standard deviations of ten types of abnormal behaviors (out of 50), among drivers participating in the study were:

1. Changing radio stations and listening to music and reading the GPS coordinates while driving (1.93 ± 1.4) (slips).

2. Overtaking a driver who drives slowly (1.90 ± 1.4, deliberate violation).

3. Unintentionally exceeding the speed limit (1.58 ± 1.3, unintentional violation).

4. Driving close to the front car and using high-beam lights as warning (1.44 ± 1.3, deliberate violation).

5. Missing a car path and having to travel a long distance to get around (1.39 ± 1.1, slips).

6. Get out of the car while the car key is still inside (1.24 ± 1.1, slips).

7. Deliberately ignoring the posted legal speed limit, late at night or early in the morning (1.22 ± 1.3, deliberate violation).

8. Choosing a wrong route to prevent being rear-ended in traffic (1.18 ± 0.9, lapse).

**Table 1. Characteristics of participant's driving behaviors, including errors, lapses, unintentional violations and deliberate violations according to their demographic variables (n = 800).**

| Variables | Categories | N (%) | Slips | Lapses | Unintentional violations | Deliberate violations |
|---|---|---|---|---|---|---|
| | | | mean (SD) | mean (SD) | mean (SD) | mean (SD) |
| **Gender** | Male | 684(85.5) | 19.88(11.62) | 7.59(5.58) | 2.93(2.34) | 16.57(13.25) |
| | Female | 116(14.5) | 17.68(10.12) | 6.29(4.81) | 2.50(2.01) | 12.58(8.49) |
| **Age** (year) | ≤ 25 | 200(25) | 23.54(13.45) | 9.05(6.86) | 3.47(2.73) | 20.49(15.19) |
| | 26–35 | 246(30.8) | 18.89(10.95) | 7.32(5.08) | 2.86(2.21) | 15.90(13.31) |
| | 36–45 | 168(21) | 17.89(9.84) | 6.92(4.80) | 2.63(1.97) | 14.27(9.78) |
| | 46–55 | 142(17.7) | 16.91(10.45) | 5.85(4.55) | 2.18(1.98) | 12.22(9.99) |
| | ≥56 | 44(5.5) | 20.11(7.98) | 7.25(4.19) | 3.34(2.15) | 14.84(9.30) |
| **Education** | ≤Diploma | 443(55.4) | 19.66(11.63) | 7.20(5.39) | 2.73(2.25) | 15.99(12.20) |
| | Associate/Bachelor | 256(32) | 19.84(11.41) | 7.59(5.81) | 3.08(2.43) | 17.00(14.34) |
| | ≥Master's Degree | 101(12.6) | 18.39(10.62) | 7.83(5.14) | 2.90(2.16) | 13.46(10.25) |
| **Driving Experience** (year) | ≤1 | 51(6.4) | 27.52(14.70) | 10.17(7.39) | 3.29(2.06) | 19.62(14.05) |
| | 2–5 | 245(30.6) | 20.49(12.63) | 7.52(6.04) | 3.10(2.68) | 17.66(15.17) |
| | 6–10 | 235(29.4) | 19.34(10.65) | 7.83(5.41) | 2.75(2.18) | 15.77(12.13) |
| | >10 | 269(33.6) | 17.39(9.35) | 6.39(4.30) | 2.68(2.05) | 13.98(10.01) |
| **Driving duration a day** (hours) | <3 | 391(48.9) | 19.16(10.61) | 7.03(4.71) | 2.85(2.22) | 14.53(10.60) |
| | 3–6 | 323(40.4) | 19.65(11.26) | 7.33(5.29) | 2.85(2.28) | 16.59(12.86) |
| | >6 | 86(10.8) | 21.03(15.11) | 9.34(8.47) | 3.00(2.74) | 20.39(18.78) |
| **Fines** | Yes | 680(85) | 20.26(11.80) | 7.81(5.65) | 3.00(2.31) | 16.72(13.19) |
| | No | 120(15) | 15.61(8.05) | 5.10(3.79) | 2.14(2.13) | 11.86(8.73) |
| **Crash** (in a year) | No crash | 115(14.4) | 18.40(9.16) | 6.86(4.84) | 2.32(2.12) | 13.89(9.77) |
| | once | 427(53.4) | 18.65(10.76) | 6.92(5.17) | 2.72(2.11) | 15.30(11.94) |
| | Twice | 135(16.9) | 19.42(10.32) | 7.74(4.87) | 2.95(2.16) | 15.75(11.49) |
| | Three times | 72(9) | 20.56(13.26) | 7.98(5.52) | 3.19(2.00) | 15.84(11.49) |
| | ≥Four times | 51(6.4) | 28.72(16.70) | 10.94(8.83) | 4.70(3.71) | 27.39(21.78) |
| **Injury severity** | No injury | 400(50) | 19.27(9.67) | 7.27(4.91) | 2.88(2.21) | 15.46(11.81) |
| | Mild injury | 200(25) | 17.61(9.63) | 6.91(4.93) | 2.51(1.95) | 15.05(11.2) |
| | Severe injury | 140(17.5) | 21.01(15.32) | 7.19(6.31) | 2.71(2.29) | 16.22(13.43) |
| | Fatal injury | 60(7.5) | 24.76(15.03) | 10.48(7.70) | 4.35(3.27) | 22.13(19.03) |
| **Total** | | | 19.56(1.14) | 7.40(5.49) | 2.87(2.30) | 15.99(1.27) |

9. Overtaking a slower vehicle over a continuous line (1.14 ± 1.4, lapse).

10. Overtaking prior to look at the mirror and hear the beep of the car behind announcing that the car in the back has already begun to overtake (1.08 ± 0.9, slips).

Table 2 shows the correlation between demographic factors and the four DBQ dimensions using the linear regression model. The results of this table show that the slips (B = -0.09, P = 0.002), lapses (B = -0.06, P = 0.004), unintentional violations (B = -0.02, P = 0.02) and deliberate violations (B = -0.26, P = 0.001) were decreased with increasing age. In women, the rates of slips (B = -3.81, P = 0.001), lapses (B = -1.88, P = 0.001), unintentional violations (B = -0.62, P = 0.001) and deliberate violations (B = -5.24, P = 0.001) were less than in men. Academic educational level had a negative effect on deliberate violations (B = -5.24, P = 0.001), a driving fine had a negative effect on slips (B = -5.09, P = 0.001), lapses (B = -2.77, P =. 0.001), unintentional violations (B = -0.92, P = 0.001) and deliberate violations (B = -5.15, P = 0.001), and driving experience had a negative effect on lapses, slips and deliberate violations (p < 0.05). Drivers who were driving more than 6 hours a day had a higher risk of lapses (B = 2.41,

**Table 2. Linear regression results between predicted values and the four DBQ factors.**

| Predictor Variables | Slips | | | Lapses | | | Unintentional violations | | | Deliberate violations | | |
|---|---|---|---|---|---|---|---|---|---|---|---|---|
| | B | Std. Error | P value | B | Std. Error | P value | B | Std. Error | P value | B | Std. Error | P value |
| **Age** (year) | -0.09 | 0.04 | 0.02 | -0.06 | 0.02 | 0.004 | -0.02 | 0.009 | 0.01 | -0.26 | 0.04 | 0.001 |
| **Gender** | -3.81 | 1.150 | 0.001 | -1.88 | 0.55 | 0.001 | -0.62 | 0.23 | 0.001 | -5.24 | 1.26 | 0.001 |
| **Education** | | | | | | | | | | | | |
| Associate/Bachelor [a] | -0.02 | 0.89 | 0.97 | 0.25 | 0.43 | 0.54 | 0.29 | 0.18 | 0.12 | 0.31 | 0.98 | 0.75 |
| ≥Master's Degree [b] | -2.33 | 1.25 | 0.06 | 0.14 | 0.60 | 0.81 | 0.009 | 0.26 | 0.97 | -4.28 | 1.37 | 0.002 |
| **Driving Experience** (year) | | | | | | | | | | | | |
| 2–5 year [c] | -7.25 | 1.70 | 0.001 | -2.74 | 0.81 | 0.001 | -0.20 | 0.35 | 0.56 | -2.06 | 1.86 | 0.26 |
| 6–10 year [d] | -8.67 | 1.75 | 0.001 | -2.54 | 0.84 | 0.003 | -0.55 | 0.36 | 0.12 | -3.56 | 1.93 | 0.06 |
| >10year [e] | -3.35 | 0.63 | 0.001 | -1.15 | 0.30 | 0.001 | -0.14 | 0.13 | 0.26 | -1.07 | 0.69 | 0.12 |
| **Driving duration a day** (Hours) | | | | | | | | | | | | |
| 3–6 [f] | 0.60 | 0.84 | 0.47 | 0.31 | 0.40 | 0.43 | 0.02 | 0.17 | 0.88 | 2.14 | 0.92 | 0.02 |
| >6 [g] | 1.97 | 1.31 | 0.13 | 2.41 | 0.63 | 0.001 | 0.19 | 0.27 | 0.47 | 6.11 | 1.44 | 0.001 |
| **Fines** | -5.09 | 1.09 | 0.001 | -2.77 | 0.52 | 0.001 | -0.92 | 0.22 | 0.001 | -5.15 | 1.20 | 0.001 |
| **Model summary** | F = 8.36 **, | | | F = 9.16 **, | | | F = 4.11 **, | | | F = 11.12 **, | | |
| | R2 = 0.09, | | | R2 = 0.10, | | | R2 = 0.05, | | | R2 = 0.12, | | |
| | Adjusted R2 = 0.08 | | | Adjusted R2 = 0.09 | | | Adjusted R2 = 0.03 | | | Adjusted R2 = 0.11 | | |

[*]: $p < 0.05$

[**]: $p < 0.01$

[***]: $p < 0.001$. B- Regression coefficient of variables in the model.

a. Education is a dummy variable taking the value 1 if the respondent is Associate/Bachelor and zero otherwise

b. Education is a dummy variable taking the value 1 if the respondent is ≥Master's Degree and zero otherwise

c. Driving experience is a dummy variable taking the value 1 if the respondent is 2-5year and zero otherwise

d. Driving experience is a dummy variable taking the value 1 if the respondent is 6-10year and zero otherwise

e. Driving experience is a dummy variable taking the value 1 if the respondent is >10 year and zero otherwise

f. Driving time is a dummy variable taking the value 1 if the respondent is 3–6 hours and zero otherwise

g. Driving time is a dummy variable taking the value 1 if the respondent is >6 hours and zero otherwise

P = 0.001) and deliberate violations (B = 6.11, P = 0.001) than drivers who were driving less than three hours a day.

The results of PO ordinal logistic regression model were presented in Table 3. Based on chi-square test ($X^2$ = 77.1, p-value<0.001) the model provides an appropriate fit. The Pseudo $R^2$ (McFadden's $R^2$) in this model was 0.14 and Nagelkerke = 0.1, suggesting that the relationship between the response variable and the predictors is small. The OR and the 95% confidence interval of the OR based on Table 3, unintentional violations and driving experience were statistically significant but other covariates are not. So for unintentional violations, one unit increases in unintentional violations, the odds of being in a lower level versus at or higher levels of crash is 1.14, given all of the other variables in the model are held constant. Also the OR of being at or below category of crash in individual with driving experience of <1 year versus individual in driving experience >10 years was 0.41, this means that the level of crash in low experience driving increases. The interpreting of threshold is as follows:

Threshold (crash = 0): This is the estimated cut point on the latent variable used to differentiate No crash from one and higher crash when values of the predictor variables are evaluated at zero. Subjects that had a value of -1.37 or less on the underlying latent variable that gave rise to our crash variable would be classified as no crash given they were for example male (when the female considered baseline) and had zero value at the covariates.

**Table 3. The multiple POM to crash level as response five ordered categories using ordered logistic regression.**

| Variables | Categories | OR* (CI 95%) | P value |
|---|---|---|---|
| lapses | | 1.01(0.98–1.01) | 0.59 |
| Slips | | 1.01(0.99–1.01) | 0.23 |
| Deliberate violations | | 1.17(0.91–1.17) | 0.22 |
| Unintentional violations | | 1.14(1.06–1.14) | <0.001 |
| Gender | Male | 1.28(0.86–1.92) | 0.22 |
| | Female (Reference category) | | |
| Age (year) | ≤ 25 | 0.97(0.46–2.05) | 0.93 |
| | 26–35 | 0.62(0.31–1.22) | 0.16 |
| | 36–45 | 1.20(0.62–2.35) | 0.59 |
| | 46–55 | 1.27(0.66–2.45) | 0.48 |
| | ≥56 (Reference category) | | |
| Education | ≤Diploma | 0.95(0.62–1.47) | 0.820 |
| | Associate/Bachelor | 0.77(0.49–1.21) | 0.255 |
| | ≥Master's Degree | | |
| Driving Experience (year) | ≤1 | 0.41(0.21–0.82) | 0.01 |
| | 2–5 | 0.82(0.54–1.25) | 0.36 |
| | 6–10 | 1.26(0.86–1.83) | 0.23 |
| | >10 (Reference category) | | |
| Driving duration a day (hours) | <3 | 0.88(0.56–1.38) | 0.57 |
| | 3–6 | 0.83(0.53–1.31) | 0.42 |
| | >6 (Reference category) | | |
| Fines | Yes | 0.91(0.62–1.34) | 0.63 |
| | No (Reference category) | | |
| Threshold Crash (in a year) | No crash | | 0.005 |
| | Once | | 0.007 |
| | Twice | | <0.001 |
| | ≥Three times | | <0.001 |

Threshold (crash = 1): This is the estimated cutpoint on the latent variable used to differentiate no and one crash from higher level of crash when values of the predictor variables are evaluated at zero. Subjects that had a value of 1.32 or greater on the underlying latent variable that gave rise to our crash variable would be classified as higher crash given they were male and had zero on other covariates. Subjects that had a value between -1.37 and 1.32 on the underlying latent variable would be classified as one crash and so on to other threshold.

In Table 4, in multinomial logistic regression model for response variable injure with 4 levels (no injury, mild injury, severe injury and fatal injury) in which "no injury" was considered as reference category. Based on chi-square test ($X^2$ = 265.5, $p_{value}$ <0.001) the model provides an appropriate fit. The Pseudo $R^2$ (McFadden's $R^2$) for this model was 0.3 and Nagelkerke = 0.12 suggesting that the relationship between the response variable and the predictors is moderate. The results showed that in Mild injury compared to no injury, the variables gender, age, education, driving experience and driving duration a day were statistically significant. In severe injury compared to no injury, gender, education, fines and driving duration were statistically significant. Also, unintentional violations, age, education, driving experience, driving duration a day were statistically significant in fatal injury compared to no injury.

**Table 4. Multiple multinomial logistic regression models using injury status as multinomial response.**

| Variables | | OR* (CI 95%) | | |
| --- | --- | --- | --- | --- |
| | | **Mild Injury VS No Injury** | **Severe Injury VS No Injury** | **Fatal Injury VS No Injury** |
| lapses | | 1.01(0.97–1.06) | 0.95(0.89–1) | 1.06(0.98–1.13) |
| Slips | | 0.98(0.96–1.01) | 1.03(1–1.06) | 1(0.96–1.04) |
| Deliberate violations | | 1.13(0.8–1.59) | 0.98(0.67–1.46) | 0.77(0.45–1.31) |
| Unintentional violations | | 0.94(0.85–1.04) | 0.97(0.87–1.09) | 1.25(1.08–1.46) * |
| Gender | Male | 3.27(1.78–6.01) * | 2.04(1.1–4.18)* | 2.89(0.92–9.03) |
| | Female (Reference category) | 1 | 1 | 1 |
| Age (year) | ≤ 25 | 0.83(0.25–2.73) | 1.23(0.4–3.77) | 0.13(0.03–0.65)* |
| | 26–35 | 1.25(0.41–3.79) | 1.07(0.39–2.93) | 0.16(0.04–0.69)* |
| | 36–45 | 2.73(0.92–8.08) | 1.8(0.68–4.8) | 0.49(0.13–1.93) |
| | 46–55 | 3.63(1.24–10.62)* | 1.72(0.65–4.52) | 0.48(0.13–1.8) |
| | ≥56 (Reference category) | 1 | 1 | 1 |
| Education | ≤Diploma | 2.32(1.19–4.5)* | 1.34(0.72–2.48) | 7.36(3.16–12.45)* |
| | Associate/Bachelor | 2.29(1.17–4.49)* | 0.29(0.14–0.62)* | 9.7(2.7–13.7)* |
| | ≥Master's Degree | 1 | 1 | 1 |
| Driving Experience (year) | ≤1 | 2.21(0.86–5.67) | 1.42(0.53–3.81) | 1.3(0.2–15.6) |
| | 2–5 | 2.86(1.62–5.04)* | 0.95(0.48–1.88) | 9.11(3.14–26.37)* |
| | 6–10 | 1.34(0.81–2.22) | 0.89(0.5–1.6) | 4.04(1.48–11.04)* |
| | >10 (Reference category) | 1 | 1 | 1 |
| Driving duration a day (hours) | <3 | 0.5(0.26,0.97)* | 0.16(0.08,0.31)* | 0.22(0.07,0.67)* |
| | 3–6 | 0.57(0.29,1.11) | 0.24(0.12,0.47)* | 0.74(0.28,1.99) |
| | >6 (Reference category) | 1 | 1 | 1 |
| Fines | Yes | 1.17(0.71,1.93) | 2.81(1.36,5.79)* | 2.29(0.72,7.27) |
| | No (Reference category) | 1 | 1 | 1 |

*statistically significant (p<0.05)

## Discussion

The results of this study showed that frequency and severity of RTIs were significantly correlated with DBQ dimensions. The frequency of abnormal driving behaviors was related to unintentional violations, deliberate violations, slip errors and lapse errors, respectively. The frequency order of DBQ dimensions in the present study differs from similar studies [30, 31]. Driving behavior is very complicated and part of the cultural behavior of individuals in societies because it is related to values, habits, attitudes and other factors [22, 29, 32–34]. Studies suggest that various internal (age, gender, cognitive, etc.) and external (social context) factors shape the driving style of individuals [21, 35]. In addition, social environment, including other road users, public social norms and formal and informal traffic laws, affects every driver; these differences in traffic culture levels are reflected in driving behaviors [29, 31, 36, 37].

Therefore, the differences in age, gender, socioeconomic and cultural contexts between the present study and similar studies may be the reason why the results of this study are inconsistent with other similar studies. On average, among Iranian drivers with respect to other Asian drivers, deliberate violations are more frequent than other violations [38]. The cultural context and formal and informal traffic laws are influential in creating an appropriate driving style and additional researches are required in this line of studies [29, 31, 36, 37].

The abnormal driving behaviors in this study, which had the highest frequency among drivers, were similar to those of other studies [30, 31, 36, 39, 40], but it was different from the

results of the studies by Gross et al. and Solman et al. [41, 42]. Azkan et al. showed that the most violations in northern European countries, known as safe driving countries (slips are more common than deliberate violations and unintentional ones), were related to highway speeds, but the most violations in southern European and Mediterranean countries were related to the deliberate and the unintentional violations; because the traffic contexts of these regions were more susceptible to interpersonal conflicts due to less developed infrastructures and lack of respect for laws and enforcement problems [22, 29].

This reflects the fact that drivers in safe countries are more aware of all their behaviors, such as unintentional violations, dangerous and illegal behaviors, due to strict law enforcement [29]. Therefore, different abnormalities and violations should be considered in different countries [22, 29].

Various studies have shown that gender and age are demographic factors correlated with DBQ dimensions [43]. These studies suggest that the odds ratio of slips, lapses, deliberate violations and unintentional violations are higher at younger ages and decreases with age [27, 44, 45]. In addition, women have less lapses and violations than men [44, 46–48]. The results also showed that age and gender had a significant influence on the odds ratio of errors and violations, and the mean all four dimensions of slip errors, lapse errors, deliberate violations and unintentional violations was higher in the less than 25 years age group compared to other age groups. In addition, the odds ratio of lapses, deliberate violations, and unintentional violations decreased with age. This may be due to the fact that young drivers accept a higher degree of risk during driving due to their over-optimism and higher driving abilities [48, 49]. Therefore, they are more at risk in traffic crash and commit more errors while driving. Furthermore, younger drivers have a higher risk of driving and see less of a risk of a traffic crash and injury [49].

However, as in other studies [14, 50, 51], the results of this study showed that in general, the severity of injury is lower in young drivers, which can be due to physiological differences, reduced cognitive and physical abilities to identify and respond appropriately in traffic emergencies with age.

There is widespread evidence that female drivers are less at risk than male drivers, and women are more cautious, precise and law-abiding than men [52, 53]. Hence, they have less odds ratio of errors and violations.

While the results of this study, like many studies [54], showed that the severity of injuries is higher in men than women. Results of the study by Russo et al. and Chen et al. showed that men suffer less severe injuries [15, 55].

Given that men are more likely than women to engage in high-risk driving behaviors such as speeding and aggressive driving, and are more likely to put themselves and other road users in dangerous situations, this difference may be due to difference between the number of men and women among participants, as 85.5% of the participants were male.

The World Health Organization states that the educational level is correlated with driving behavior [56]. Overall, previous studies showed that drivers' level of education do not increase safety of their driving [57, 58]. The results of the present study showed that educational level was only correlated with deliberate violations. Similar to the results of other studies conducted in Iran, these findings suggest that, despite considerable variation, Iranians behave the same in their approach to driving, especially in the cultural context. This may be due to the lack of Iranian public awareness regarding socioeconomic situation and traffic [59]. Besides, this may be due to the fact that traffic laws and their applications in Iran are not standardized compared to Western countries.

In addition, there are driving norms (real norms) in Iran that are quite different from the norms taught in driving classes (ideal driving norms), and what governs in practice, not the

ideal driving norms, as the real norms are accepted by the society at large with any education. Therefore, if one is driving based on patterns conflicting with actual driving norms followed by other drivers (such as not exceeding the speed limit by drivers, unauthorized overtaking, no excessive change of direction, lack of no entry road markings, and lack of no running from red light) disrupts the driving order (order based on real norms), causing other drivers to become angry and reprimanded in various ways (humiliation, derision and insult). On the other hand, because drivers with high educational levels have a rational understanding of environmental conditions and risk factors and pay more attention to traffic signs and barriers [60], deliberate violations and severity of RTIs are less likely to occur. These results are consistent with the findings of similar studies [27, 61].

On the other hand, studies have shown that with increasing level of education, a person's attention to the impact of high-risk behaviors (such as speeding) on RTIs and severity of RTIs will increase. Educated people also wear seat belts because of improving their perceived safety value, to avoid being harmed. In addition, educated people usually use safer and higher quality vehicles due to their better socio-economic status [62].

Circadian driving hours predicted deliberate violations and lapses, which is consistent with the results of similar studies [27, 63], which is justified by fatigue induced by driving [24].

The motivation for driving while fatigued may be due to overestimation of capabilities and miscalculation of the cost of outcome by the drivers [64]. The results of several studies, similar to the present study, show that the driving experience is correlated with DBQ dimensions [27, 63, 65–67]. This may be because the driving experience is correlated with anger and violent driving behavior, thereby affecting driving behavior [66, 68]. The results of several studies, similar to the present study, show that the driving experience is correlated with DBQ dimensions [27, 63, 65–67]. This may be because the driving experience is correlated with anger and violent driving behavior, thereby affecting driving behavior [66, 68]. On the other hand, driving skills, including information processing and motor skills, improve with practice and increased driving experience. It affects the drivers' self-esteem as well as their behaviors [69].

Evidence indicates an increase in aberrant driving behaviors and traffic offences during within the first three years after licensure [70]. Driving experience can significantly affect, drivers' perception of risk [71]. Experienced drivers can have much more knowledge and skills to drive safely and appropriately in different traffic conditions.

The results of this study showed that high experienced drivers (>10 years) have a reduced probability of severe and fatal injuries compared with mid experienced drivers (2–5 years of driving experience). This result may be attributed to the improved driving skills and handling capabilities in traffic emergencies, among high experienced drivers [72].

The effect of driving experience on driver injury severity indicated that drivers with mid experience should be continuously educated to reduce the severity of road traffic injuries.

The current study had following limitations. In this study, self-report DBQ measurement had no data on observations. However, various studies have shown that self-report driving behavior is consistent with actual performance [44, 73]. On the other hand, cross-sectional design of this study does not determine the causal correlation between DBQ dimensions with frequency of traffic crashes and injuries and severity of RTIs. However, cross-sectional studies are an important tool for identifying risk indices and safeguards for future longitudinal assessments. Providing acceptable social responses are another limitation of this study. Stutts et al. found that acceptable DBQ responsiveness was low given the rate of traffic crashes, although drivers may have overlooked or underestimated the number of traffic crashes encountered during the years of driving [19].

This study has positive aspects reinforcing the findings. In the present study, a relatively large sample size of the drivers was investigated. The sample, selected by a three-stage random

strategy, included drivers in different parts of the city. Strong sampling and high power of generalize ability are the strengths of this study. Much of the previous research on driving behavior has focused on examining the DBQ dimensions of most professional drivers such as taxi and private drivers of passenger terminals and travel agents, city bus drivers and truck drivers [30, 31, 39, 41, 42, 74]. In addition, there is limited evidence to examine a wide range of demographic variables, social characteristics and DBQ dimensions and about RTIs among public drivers in the community.

## Conclusion

According to the results of the present study, use of driving behavior questionnaires can be a satisfactory tool for studying the behavior of drivers in different groups focusing on abnormal behaviors and traffic crashes.

In addition, odds ratio of errors and violations was higher in younger male drivers, and drivers with more crashes had greater errors and violations. Therefore, policymaking and system organizing in relation to traffic injury prevention seems to be a health priority and a key tool in promoting safety. By studying drivers and changing their behaviors, the odds ratio of abnormal behavior and thus other factors affecting traffic crashes and resulting injuries and negative effects can be mitigated in the society. Eventually, findings of this study can be utilized to develop safe driving behavior interventions among the drivers.

## Supporting information

**S1 Data.**
(SAV)

## Acknowledgments

The authors thank the Deputy of Research and Technology of Hamadan University of Medical Sciences who approved this study (number: 930126254). The authors also thank all the drivers who participated in this study.

## Author Contributions

**Conceptualization:** Forouzan Rezapur-Shahkolai, Tahereh Etesamifard, Samane Shirahmadi.

**Data curation:** Samane Shirahmadi.

**Formal analysis:** Tahereh Etesamifard, Ghodratollah Roshanaei, Samane Shirahmadi.

**Funding acquisition:** Forouzan Rezapur-Shahkolai.

**Investigation:** Malihe Taheri, Tahereh Etesamifard.

**Methodology:** Forouzan Rezapur-Shahkolai, Malihe Taheri, Tahereh Etesamifard, Ghodratollah Roshanaei, Samane Shirahmadi.

**Project administration:** Forouzan Rezapur-Shahkolai, Samane Shirahmadi.

**Resources:** Malihe Taheri, Tahereh Etesamifard, Ghodratollah Roshanaei.

**Software:** Tahereh Etesamifard, Ghodratollah Roshanaei.

**Supervision:** Forouzan Rezapur-Shahkolai, Samane Shirahmadi.

**Writing – original draft:** Samane Shirahmadi.

**Writing – review & editing:** Forouzan Rezapur-Shahkolai, Malihe Taheri, Tahereh Etesami-fard, Ghodratollah Roshanaei, Samane Shirahmadi.

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
