## [Decision Letter · Decision Letter 0]

9 Jun 2020

PONE-D-20-13494

Dimensions of aberrant driving behaviors and their association with road traffic injuries among drivers

PLOS ONE

Dear Dr. Shirahmadi,

Thank you for submitting your manuscript to PLOS ONE. After careful consideration, we feel that it has merit but does not fully meet PLOS ONE’s publication criteria as it currently stands. Therefore, we invite you to submit a revised version of the manuscript that addresses the points raised during the review process.

We look forward to receiving your revised manuscript.

Kind regards,

Feng Chen

Academic Editor

PLOS ONE

Journal Requirements:

Reviewers' comments:

Reviewer's Responses to Questions

**Comments to the Author**

1. Is the manuscript technically sound, and do the data support the conclusions?

Reviewer #1: Yes

Reviewer #2: Partly

2. Has the statistical analysis been performed appropriately and rigorously? 

Reviewer #1: No

Reviewer #2: Yes

3. Have the authors made all data underlying the findings in their manuscript fully available?

Reviewer #1: Yes

Reviewer #2: Yes

4. Is the manuscript presented in an intelligible fashion and written in standard English?

Reviewer #1: Yes

Reviewer #2: Yes

5. Review Comments to the Author

Reviewer #1: The paper investigates the dimensions of aberrant driving behaviors and their effects on the crash frequency and severity of drivers in Iran. The research topic is interesting and worth of investigation. However, the Pearson tests and logit regressions used are somewhat simple. More works on traffic frequency and severity modeling should be referred to and acknowledged in the Introduction Section, including:

Investigating the impacts of real-time weather conditions on freeway crash severity: A Bayesian spatial analysis. International Journal of Environmental Research and Public Health, 2020, 17(8), 2768.

Spatial joint analysis for zonal daytime and nighttime crash frequencies using a Bayesian bivariate conditional autoregressive model. Journal of Transportation Safety and Security, 2020, 12(4): 566-585.

Jointly modeling area-level crash rates by severity: A Bayesian multivariate random-parameters spatio-temporal Tobit regression. Transportmetrica A: Transport Science, 2019, 15(2): 1867-1884.

According to the description in the Sampling Section, the drivers selected can be grouped by geographical districts, which may result in spatial correlation across adjacent drivers. Accounting for the spatial correlation in the regression models would improve their estimation capacities and reduce model misspecification, as indicated in the following papers.

Reviewer #2: The topic of this paper is interesting and the results are meaningful and useful. There are several suggestions to improve this paper.

1. In the abstract, “to develop safe driving behavior interventions” could be “to develop driving behavior interventions”.

2. “According to a report by the Traffic Police of Iran in 2014, human behavior (71%), technical defect (18%) and traffic engineering (11%) were effective in the traffic crashes (12).” The word effective is misleading here.

3. “et al” in several places of this paper is a typo.

4. The Pearson tests and logit regressions used in this paper are somehow a bit simple. Some papers on traffic safety modeling and experiments could be referred to and acknowledged in the Introduction. For example, the following ones.

[1] Examining the safety of trucks under crosswind at bridge-tunnel section: A driving simulator study, Tunnelling and Underground Space Technology, 2019, 92, 103034. https://doi.org/10.1016/j.tust.2019.103034

[2] Chen, Feng; Chen, Suren; Ma, Xiaoxiang. Analysis of hourly crash likelihood using unbalanced panel data mixed logit model and real-time driving environmental big data. 2018, JOURNAL OF SAFETY RESEARCH. 65: 153-159.

[3] Investigation on the Injury Severity of Drivers in Rear-End Collisions Between Cars Using a Random Parameters Bivariate Ordered Probit Model, International Journal of Environmental Research and Public Health, 2019, 16(14) , 2632.

[4] Investigating the Differences of Single- and Multi-vehicle Accident Probability Using Mixed Logit Model, Journal of Advanced Transportation, 2018, UNSP 2702360.

6. PLOS authors have the option to publish the peer review history of their article (what does this mean?). If published, this will include your full peer review and any attached files.

Reviewer #1: No

Reviewer #2: No

---

## [Author Response · Author response to Decision Letter 0]

5 Aug 2020

Author’s response to reviews

Title: Dimensions of aberrant driving behaviors and their association with road traffic injuries among drivers 

Authors:

Forouzan Rezapur-Shahkolai (forouzan.rezapour@gmail.com)

Malihe Taheri (ma.taheri@umsha.ac.ir) 

Tahereh Etesamifard (tahere_health@rocketmail.com)

Ghodratollah Roshanaei (gh.roshanaei@umsha.ac.ir)

Samaneh Shirahmadi (shirahmadi_s@yahoo.com)

Version: 1 Date: 2 August 2020

Author's response to reviews: see over

We thank all the reviewers for their valuable feedback and taking the time to provide useful comments which improved our manuscript entitled “Dimensions of aberrant driving behaviors and their association with road traffic injuries among drivers”. 

Based on the constructive comments the following changes have been made. 

Response to Reviewer 1:

1. The research topic is interesting and worth of investigation. However, the Pearson tests and logit regressions used are somewhat simple. 

Response: Thank you for your valuable comment. We changed our analysis and from ordinal logistic regression model (proportional odds model (POM)) and the chi-squared score test for the proportional odds assumption in ordinal logistic was employed to see whether the main model assumption was violated or not. Also, multinomial logistic regression (multinomial regression) was used to predict a nominal dependent variable (injury variable as multinomial variable) given one or more independent variables.

- In the abstract section:

- We have deleted these sentences from abstract section: " Data were statistically analyzed using numerical indices, linear regression analysis, multiple regression analysis and Pearson correlation.", "Traffic crashes and the severity of related injuries were significantly correlated with DBQ dimensions" and added this sentences in this section: " Data were statistically analyzed using numerical indices, linear regression analysis, Pearson correlation, ordinal logistic regression model and multinomial logistic regression." and "Age, gender, educational level, driving experience and driving hours during the day were significantly associated with DBQ dimensions and severity of road traffic injuries.

- In the method section:

- We have deleted these sentences from data analysis section" In addition, due to the correlation between DBQ dimensions and the rate of traffic crashes and the resulting injuries in different studies, multiple-logistic regression model was performed to investigate this correlation.", and added this paragraphs

- "Analysis of crash variable as ordinal response variable was performed using ordinal logistic regression model (proportional odds model (POM)). To estimate the Ln (odds) of being at or below the jth category, the PO model can be rewritten as:

Thus, this model predicts cumulative logits J-1 response categories. By transforming the cumulative logits, we can obtain the estimated cumulative odds as well as the cumulative probabilities being at or below the jth category. The chi-squared score test for the proportional odds assumption in ordinal logistic was employed to see whether the main model assumption was violated or not. In this study ordered dependent variable consists of 5 ordinal categories (no traffic crash, one traffic crash, two traffic crashes, three traffic crashes, four or more traffic crashes). Also, multinomial logistic regression (multinomial regression) was used to predict a multinomial dependent variable (injury variable) given one or more independent variables. The 95% confidence interval (CI) and odds ratio were calculated. A Significance level of P <0.05 was considered in all statistical tests.

In the result section:

- We deleted Tables 2 and 4 and added two more tables.

- We have added these paragraphs in the result section:

"The results of PO ordinal logistic regression model were presented in Table 3. Based on chi-square test (X2=77.1, p-value<0.001) the model provides an appropriate fit. The Pseudo R2 (McFadden’s R2) in this model was 0.14 and Nagelkerke=0.1, suggesting that the relationship between the response variable and the predictors is small. The OR and the 95% confidence interval of the OR based on Table 3, unintentional violations and driving experience were statistically significant but other covariates are not. So for unintentional violations, one unit increases in unintentional violations, the odds of being in a lower level versus at or higher levels of crash is 1.14, given all of the other variables in the model are held constant. Also the OR of being at or below category of crash in individual with driving experience of <1 year versus individual in driving experience >10 years was 0.41, this means that the level of crash in low experience driving increases. The interpreting of threshold is as follows:

Threshold (crash=0): This is the estimated cut point on the latent variable used to differentiate No crash from one and higher crash when values of the predictor variables are evaluated at zero. Subjects that had a value of -1.37 or less on the underlying latent variable that gave rise to our crash variable would be classified as no crash given they were for example male (when the female considered baseline) and had zero value at the covariates.

Threshold (crash=1): This is the estimated cut point on the latent variable used to differentiate no and one crash from higher level of crash when values of the predictor variables are evaluated at zero. Subjects that had a value of 1.32 or greater on the underlying latent variable that gave rise to our crash variable would be classified as higher crash given they were male and had zero on other covariates. Subjects that had a value between -1.37 and 1.32 on the underlying latent variable would be classified as one crash and so on to other threshold."

And" In table 4, in multinomial logistic regression model for response variable injure with 4 levels (no injury, mild injury, severe injury and fatal injury) in which “no injury” was considered as reference category. Based on chi-square test (X2=265.5, pvalue<0.001) the model provides an appropriate fit. The Pseudo R2 (McFadden’s R2) for this model was 0.3 and Nagelkerke=0.12 suggesting that the relationship between the response variable and the predictors is moderate. The results showed that in Mild injury compared to no injury, the variables gender, age, education, driving experience and driving duration a day were statistically significant. In severe injury compared to no injury, gender, education, fines and driving duration were statistically significant. Also, unintentional violations, age, education, driving experience, driving duration a day were statistically significant in fatal injury compared to no injury."

In the discussion section:

- We have added these paragraphs in discussion section:

- “However, as in other studies the results of this study showed that

in general, the severity of injury is lower in young drivers, which can be due to physiological differences, reduced cognitive and physical abilities to identify and respond appropriately in traffic emergencies with age."

- “While the results of this study, like many studies, showed that the severity of injuries is higher in men than women. Results of the study by Russo et al. and Chen et al. showed that men suffer less severe injuries. Given that men are more likely than women to engage in high-risk driving behaviors such as speeding and aggressive driving, and are more likely to put themselves and other road users in dangerous situations, this difference may be due to difference between the number of men and women among participants, as 85.5% of the participants were male."

- “On the other hand, studies have shown that with increasing level of education, a person's attention to the impact of high-risk behaviors (such as speeding) on RTIs and severity of RTIs will increase. Educated people also wear seat belts because of improving their perceived safety value, to avoid being harmed. In addition, educated people usually use safer and higher quality vehicles due to their better socio-economic status."

- “The motivation for driving while fatigued may be due to overestimation of capabilities and miscalculation of the cost of outcome by the drivers."

- “Evidence indicates an increase in aberrant driving behaviors and traffic offences during within the first three years after licensure. Driving experience can significantly affect drivers' perception of risk. Experienced drivers can have much more knowledge and skills to drive safely and appropriately in different traffic conditions.

- The results of this study showed that high experienced drivers (>10 years) have a reduced probability of severe and fatal injuries compared with mid experienced drivers (2-5 years of driving experience). This result may be attributed to the improved driving skills and handling capabilities in traffic emergencies, among high experienced drivers. The effect of driving experience on driver injury severity indicated that drivers with mid experience should be continuously educated to reduce the severity of road traffic injuries."

2. More works on traffic frequency and severity modeling should be referred to and acknowledged in the Introduction Section, including:

Investigating the impacts of real-time weather conditions on freeway crash severity: A Bayesian spatial analysis. International Journal of Environmental Research and Public Health, 2020, 17(8), 2768.

Spatial joint analysis for zonal daytime and nighttime crash frequencies using a Bayesian bivariate conditional autoregressive model. Journal of Transportation Safety and Security, 2020, 12(4): 566-585.

Jointly modeling area-level crash rates by severity: A Bayesian multivariate random-parameters spatio-temporal to bit regression. Transportmetrica A: Transport Science, 2019, 15(2): 1867-1884.

Response: Thank you for your valuable comment. We have added some studies based traffic frequency and severity modeling in introduction, as recommended. References # 11-13.

3. According to the description in the Sampling Section, the drivers selected can be grouped by geographical districts, which may result in spatial correlation across adjacent drivers. Accounting for the spatial correlation in the regression models would improve their estimation capacities and reduce model misspecification, as indicated in the following papers.

Response: Thank you for your valuable comment. GLMM model in this study is not appropriate because: Random effect model was used to model of correlated or clustered data (in longitudinal as well as matched set of observation). But in present study, the data was collected from cross-sectional study as random sample of population. This type of data is independent and it is no need to use the mixed effect model to analyze them.

Response to Reviewer 2:

1. In the abstract, “to develop safe driving behavior interventions” could be “to develop driving behavior interventions”.

Response: Thank you for your valuable comment. The sentence has been revised as: “The findings of this study can be utilized to develop driving behavior interventions among the drivers”. Page 3 line 60.

2. “According to a report by the Traffic Police of Iran in 2014, human behavior (71%), technical defect (18%) and traffic engineering (11%) were effective in the traffic crashes (12).” The word effective is misleading here.

Response: We have deleted this word. Page 5, line 107.

3. “et al” in several places of this paper is a typo.

Response: We have revised this word throughout the manuscript.

4. The Pearson tests and logit regressions used in this paper are somehow a bit simple. 

Response: Thank you for your valuable comment. We changed our analysis and from ordinal logistic regression model (proportional odds model (POM)) and the chi-squared score test for the proportional odds assumption in ordinal logistic was employed to see whether the main model assumption was violated or not. Also, multinomial logistic regression (multinomial regression) was used to predict a nominal dependent variable (injury variable as multinomial variable) given one or more independent variables.

- In the abstract section:

- We have deleted these sentences from abstract section" Data were statistically analyzed using numerical indices, linear regression analysis, multiple regression analysis and Pearson correlation.", "Traffic crashes and the severity of related injuries were significantly correlated with DBQ dimensions" and added this sentences in this section: " Data were statistically analyzed using numerical indices, linear regression analysis, Pearson correlation, ordinal logistic regression model and multinomial logistic regression." And "Age, gender, educational level, driving experience and driving hours during the day were significantly associated with DBQ dimensions and severity of road traffic injuries.

- In the method section:

- We have deleted these sentences from data analysis section" In addition, due to the correlation between DBQ dimensions and the rate of traffic crashes and the resulting injuries in different studies, multiple-logistic regression model was performed to investigate this correlation.", and added this paragraphs

- "Analysis of crash variable as ordinal response variable was performed using ordinal logistic regression model (proportional odds model (POM)). To estimate the Ln (odds) of being at or below the jth category, the PO model can be rewritten as:

Thus, this model predicts cumulative logits J-1 response categories. By transforming the cumulative logits, we can obtain the estimated cumulative odds as well as the cumulative probabilities being at or below the jth category. The chi-squared score test for the proportional odds assumption in ordinal logistic was employed to see whether the main model assumption was violated or not.In this study ordered dependent variable consists of 5 ordinal categories (no traffic crash, one traffic crash, two traffic crashes, three traffic crashes, four or more traffic crashes). Also, multinomial logistic regression (multinomial regression) was used to predict a multinomial dependent variable (injury variable) given one or more independent variables. The 95% confidence interval (CI) and odds ratio were calculated. A Significance level of P <0.05 was considered in all statistical tests.

- In the result section:

- We deleted Tables 2 and 4 and added two more tables.

- We have added these paragraphs in result section:

- "The results of PO ordinal logistic regression model were presented in Table 3. Based on chi-square test (X2=77.1, p-value<0.001) the model provides an appropriate fit. The Pseudo R2 (McFadden’s R2) in this model was 0.14 and Nagelkerke=0.1, suggesting that the relationship between the response variable and the predictors is small. The OR and the 95% confidence interval of the OR based on Table 3, unintentional violations and driving experience. were statistically significant but other covariates are not. So for unintentional violations, one unit increases in unintentional violations, the odds of being in a lower level versus at or higher levels of crash is 1.14, given all of the other variables in the model are held constant. Also the OR of being at or below category of crash in individual with driving experience of <1 year versus individual in driving experience >10 years was 0.41, this means that the level of crash in low experience driving increases. The interpreting of threshold is as follows:

Threshold (crash=0): This is the estimated cut point on the latent variable used to differentiate No crash from one and higher crash when values of the predictor variables are evaluated at zero. Subjects that had a value of -1.37 or less on the underlying latent variable that gave rise to our crash variable would be classified as no crash given they were for example male (when the female considered baseline) and had zero value at the covariates.

Threshold (crash=1): This is the estimated cut point on the latent variable used to differentiate no and one crash from higher level of crash when values of the predictor variables are evaluated at zero. Subjects that had a value of 1.32 or greater on the underlying latent variable that gave rise to our crash variable would be classified as higher crash given they were male and had zero on other covariates. Subjects that had a value between -1.37 and 1.32 on the underlying latent variable would be classified as one crash and so on to other threshold."

And" In table 4, in multinomial logistic regression model for response variable injure with 4 levels (no injury, mild injury, severe injury and fatal injury) in which “no injury” was considered as reference category. Based on chi-square test (X2=265.5, pvalue<0.001) the model provides an appropriate fit. The Pseudo R2 (McFadden’s R2) for this model was 0.3 and Nagelkerke=0.12 suggesting that the relationship between the response variable and the predictors is moderate. The results showed that in Mild injury compared to no injury, the variables gender, age, education, driving experience and driving duration a day were statistically significant. In severe injury compared to no injury, gender, education, fines and driving duration were statistically significant. Also Unintentional violations, Age, education, driving experience, driving duration a day were statistically significant in fatal injury compared to no injury."

- In the discussion section:

- We have added these paragraphs in discussion section:

- “However, as in other studies the results of this study showed that

in general, the severity of injury is lower in young drivers, which can be due to physiological differences, reduced cognitive and physical abilities to identify and respond appropriately in traffic emergencies with age."

- “While the results of this study, like many studies, showed that the severity of injuries is higher in men than women. Results of the study by Russo et al. and Chen et al. showed that men suffer less severe injuries. Given that men are more likely than women to engage in high-risk driving behaviors such as speeding and aggressive driving, and are more likely to put themselves and other road users in dangerous situations, this difference may be due to difference between the number of men and women among participants, as 85.5% of the participants were male."

- “On the other hand, studies have shown that with increasing level of education, a person's attention to the impact of high-risk behaviors (such as speeding) on RTIs and severity of RTIs will increase. Educated people also wear seat belts because of improving their perceived safety value, to avoid being harmed. In addition, educated people usually use safer and higher quality vehicles due to their better socio-economic status."

- “The motivation for driving while fatigued may be due to overestimation of capabilities and miscalculation of the cost of outcome by the drivers."

- “Evidence indicates an increase in aberrant driving behaviors and traffic offences during within the first three years after licensure. Driving experience can significantly affect drivers' perception of risk. Experienced drivers can have much more knowledge and skills to drive safely and appropriately in different traffic conditions.

- The results of this study showed that high experienced drivers (>10 years) have a reduced probability of severe and fatal injuries compared with mid experienced drivers (2-5 years of driving experience). This result may be attributed to the improved driving skills and handling capabilities in traffic emergencies, among high experienced drivers. The effect of driving experience on driver injury severity indicated that drivers with mid experience should be continuously educated to reduce the severity of road traffic injuries."

Some papers on traffic safety modeling and experiments could be referred to and acknowledged in the Introduction. For example, the following ones.

[1] Examining the safety of trucks under crosswind at bridge-tunnel section: A driving simulator study, Tunnelling and Underground Space Technology, 2019, 92, 103034. https://doi.org/10.1016/j.tust.2019.103034

[2] Chen, Feng; Chen, Suren; Ma, Xiaoxiang. Analysis of hourly crash likelihood using unbalanced panel data mixed logit model and real-time driving environmental big data. 2018, JOURNAL OF SAFETY RESEARCH. 65: 153-159.

[3] Investigation on the Injury Severity of Drivers in Rear-End Collisions Between Cars Using a Random Parameters Bivariate Ordered Probit Model, International Journal of Environmental Research and Public Health, 2019, 16(14) , 2632.

[4] Investigating the Differences of Single- and Multi-vehicle Accident Probability Using Mixed Logit Model, Journal of Advanced Transportation, 2018, UNSP 2702360. 

Response: Thank you for your valuable comment. We have added some studies based traffic frequency and severity modeling in introduction, as recommended. References # 14-15.

---

## [Decision Letter · Decision Letter 1]

24 Aug 2020

Dimensions of aberrant driving behaviors and their association with road traffic injuries among drivers

PONE-D-20-13494R1

Dear Dr. Shirahmadi,

We’re pleased to inform you that your manuscript has been judged scientifically suitable for publication and will be formally accepted for publication once it meets all outstanding technical requirements.

Kind regards,

Feng Chen

Academic Editor

PLOS ONE

Additional Editor Comments (optional):

Reviewers' comments:

Reviewer's Responses to Questions

**Comments to the Author**

1. If the authors have adequately addressed your comments raised in a previous round of review and you feel that this manuscript is now acceptable for publication, you may indicate that here to bypass the “Comments to the Author” section, enter your conflict of interest statement in the “Confidential to Editor” section, and submit your "Accept" recommendation.

Reviewer #1: All comments have been addressed

Reviewer #2: All comments have been addressed

2. Is the manuscript technically sound, and do the data support the conclusions?

Reviewer #1: (No Response)

Reviewer #2: Yes

3. Has the statistical analysis been performed appropriately and rigorously? 

Reviewer #1: (No Response)

Reviewer #2: Yes

4. Have the authors made all data underlying the findings in their manuscript fully available?

Reviewer #1: (No Response)

Reviewer #2: Yes

5. Is the manuscript presented in an intelligible fashion and written in standard English?

Reviewer #1: (No Response)

Reviewer #2: Yes

6. Review Comments to the Author

Reviewer #1: (No Response)

Reviewer #2: (No Response)

7. PLOS authors have the option to publish the peer review history of their article (what does this mean?). If published, this will include your full peer review and any attached files.

Reviewer #1: No

Reviewer #2: No

---

## [Editor Report · Acceptance letter]

31 Aug 2020

PONE-D-20-13494R1 

Dimensions of aberrant driving behaviors and their association with road traffic injuries among drivers 

Dear Dr. Shirahmadi:

I'm pleased to inform you that your manuscript has been deemed suitable for publication in PLOS ONE. Congratulations! Your manuscript is now with our production department. 

Kind regards, 

on behalf of

Dr. Feng Chen 

Academic Editor

PLOS ONE